# HIV RNA/DNA Levels at Diagnosis Can Predict Immune Reconstitution: A Longitudinal Analysis

**DOI:** 10.3390/microorganisms11061510

**Published:** 2023-06-06

**Authors:** Dimitrios Basoulis, Nikos Pantazis, Dimitrios Paraskevis, Panos Iliopoulos, Martha Papadopoulou, Karolina Akinosoglou, Angelos Hatzakis, George L. Daikos, Mina Psichogiou

**Affiliations:** 11st Department of Internal Medicine, Laiko General Hospital, Medical School, National and Kapodistrian University of Athens, 11527 Athens, Greece; papamartha90@gmail.com (M.P.); gldaikos@gmail.com (G.L.D.); 2Department of Hygiene, Epidemiology and Medical Statistics, Medical School, National and Kapodistrian University of Athens, 11527 Athens, Greece; npantaz@med.uoa.gr (N.P.); dparask@med.uoa.gr (D.P.); panos_iliopoulos@hotmail.com (P.I.); ahatzak@med.uoa.gr (A.H.); 3Department of Internal Medicine and Infectious Diseases, University General Hospital of Patras, Medical School, University of Patras, 26504 Patras, Greece; akin@upatras.gr

**Keywords:** HIV DNA, immune reconstitution, viral latency

## Abstract

Background: HIV DNA mirrors the number of infected cells and the size of the HIV viral reservoir. The aim of this study was to evaluate the effect of pre-cART HIV DNA levels as a predictive marker of immune reconstitution and on the post-cART CD4 counts trends. Methods: HIV DNA was isolated from PBMCs and quantified by real-time PCR. Immune reconstitution was assessed up to four years. Piecewise-linear mixed models were used to describe CD4 count changes. Results: 148 people living with HIV (PLWH) were included. The highest rate of immune reconstitution was observed during the first trimester. There was a trend showing that high HIV RNA level resulted in greater increase in CD4 count, especially during the first trimester of cART (difference above vs. below median 15.1 cells/μL/month; 95% CI −1.4–31.5; *p* = 0.073). Likewise, higher HIV DNA level would predict greater CD4 increases, especially after the first trimester (difference above vs. below median 1.2 cells/μL/month; 95% CI −0.1–2.6; *p* = 0.071). Higher DNA and RNA levels combined were significantly associated with greater CD4 increase past the first trimester (difference high/high vs. low/low 2.1 cells/μL/month; 95% CI 0.3–4.0; *p* = 0.024). In multivariable analysis, lower baseline CD4 counts predicted a greater CD4 rise. Conclusions: In successfully treated PLWH, pre-cART HIV DNA and HIV RNA levels are predictors of immune reconstitution.

## 1. Introduction

Human Immunodeficiency Virus type 1 (HIV-1) is a retrovirus bearing two copies of a single-chain RNA molecule. During the natural course of human infection, the viral particle binds to the target host cell via its envelope glycoproteins gp120 and gp41 attaching to the CD4 surface protein of T lymphocytes [1]. The viral envelope fuses with the cell membrane and the virus core enters the cytoplasm. The core proceeds to uncoat, releasing the viral RNA, which then it is transcribed into proviral DNA by the reverse transcriptase. Proviral DNA is then transported inside the cell’s nucleus and using another HIV-encoded enzyme, the integrase, it becomes integrated into the host cell’s DNA structure [1]. Monocytes/macrophages, microglial cells, and latently infected CD4 T-cells contain integrated provirus DNA, essentially becoming a long-living reservoir for HIV [2]. These infected cells are not impeded in their normal functions by the latent integrated virus and serve as a long-term viral storage site that can successfully evade the immune system.

HIV DNA is a marker that provides an estimate of the number of infected cells and on the size of the latent viral reservoir which is established shortly after primary HIV infection. Research has shown that it is strongly predictive of HIV disease progression, in the absence of combination anti-retroviral therapy (cART), a finding that is independent of plasma HIV RNA levels or CD4 cell counts [3,4,5]. The highest levels have been observed in patients with therapeutic failure [6] and are associated with faster progression to AIDS and an increased risk of death while the lowest levels have been noted in long term non-progressors and elite controllers [7,8]. Greater HIV DNA levels have also been associated with increased mortality; patients with >3 log_10_ DNA loads were two times likelier to die of any causes regardless of treatment in a recent west-African trial [9]. Among patients receiving antiviral treatment, an association has also been demonstrated between greater HIV DNA levels and treatment failure after treatment initiation [3]. In patients with advanced therapeutic failure, greater DNA levels were associated with lower CD4 gains [6]. Measurements of HIV RNA levels are part of the standard of care for people living with HIV (PLWH). Greater RNA levels have been associated with poor response to treatment and slower increases in CD4 counts [10].

On the other hand, CD4 cell count is a major marker for initial clinical assessment, and it is used for monitoring therapeutic efficacy [11,12]. Despite the low mortality rate in patients on cART with a sustained virological response, a higher CD4 cell count is always associated with a reduced risk of a new AIDS event or death [13]. However, despite CD4 cell count increase and treatment, some patients fail to exhibit satisfactory immune reconstitution. Results from the ETOILE study have shown that in patients with treatment failure, greater levels of HIV DNA, reflecting a larger latent viral reservoir in the host’s cells, were associated with smaller CD4 gains [6].

The aim of the study was to evaluate the impact of pre-treatment HIV DNA levels as an early predictive marker of immune reconstitution in patients receiving cART and having sustained virological response and to assess the effects of pre-cART HIV DNA levels on the post-cART CD4 cell counts trends.

Given the significance of HIV DNA as a prognostic marker for disease progression and response to treatment, assessing the effect of pre-cART HIV DNA on immunological recovery will provide additional information regarding the prediction of immune restoration. Combined use of prognostic markers should prove useful in individualized patient management and a helpful guide charting therapeutic strategy.

## 2. Materials and Methods

### 2.1. Study Design

This is a retrospective cohort study of HIV-infected patients followed in First Dept of Internal Medicine, Laiko General Hospital, National and Kapodistrian University of Athens. Cryopreserved lymphocyte samples before cART initiation were routinely collected.

All newly diagnosed patients who had initiated cART within the HIV-infected cohort of our clinic with at least one cryopreserved peripheral blood mononuclear cell (PBMC) sample available before cART, who had been followed for at least two years after cART initiation and had achieved sustained virological response, were eligible for this study.

Type of treatment was distributed as a protease inhibitor-based (PI), non-nucleoside reverse transcriptase inhibitor-based (NNRTI), or an integrase inhibitor-based regimens (INSTI) in combination with two nucleoside or nucleotide reverse transcriptase inhibitors (NRTI). Sustained virological response was defined as viral load lower than 50 copies/mL in two consecutive measurements after cART initiation retained throughout the follow up.

Other covariates included: baseline age, gender, transmission group, AIDS status (defined as either having a diagnosis of an AIDS defining condition and/or a CD4 count < 200 cells/μL), pre-cART HIV RNA levels in plasma, the baseline, pre-cART and nadir CD4 cell count and the cART regimen. Demographics, consecutive HIV RNA measurements and CD4 cell counts were extracted from electronic patient records.

Finally, we included data pertaining to cART complications after initiation, i.e., immune reconstitution inflammatory syndrome and mortality for those participants that have remained in long-term care.

### 2.2. Sample Size Estimation

The sample size calculation was performed using a method based on simulations [14]. The evolution of CD4 cell counts over time was assumed to follow a piecewise linear mixed model, in the square root scale, with random intercept and random slopes.

The overall (i.e., irrespectively of HIV DNA levels) average rate of CD4 cell count increase was assumed to be 1.46 (cells/μL)^1/2^ per month for the first 3 months and 0.17 (cells/μL)^1/2^ per month for the remaining 21 months. Average CD4 cell count at cART initiation in the square root scale was assumed to be 15.65 (cells/μL)^1/2^. These values correspond to starting cART with a CD4 count of 245 cells/μL and reaching 559 cells/μL after 2 years of treatment. All parameter values were based on the analysis of post-cART CD4 cell count measurements, censored at 2 years after cART initiation, derived from the Athens Multicenter AIDS Cohort Study (AMACS) collaboration [12], using an equivalent model (unpublished results).

HIV DNA was assumed, based on previous work, to be distributed normally in the log_10_ scale with mean (SD) 2.87 (0.67) log_10_ copies/mL [15]. It was assumed that HIV DNA affects both slopes in a linear way such that those with 1 log_10_ higher pre cART HIV DNA have ≈23% slower CD4 count increases (in the square root scale). To illustrate the strength of this association, a 20% decrease in the CD4 slopes is expected for individuals having HIV DNA levels above the median of its distribution compared to those having HIV DNA levels below the median value. We allowed for a 5% yearly rate of, completely at random, drop-out.

The simulated data were analyzed using a piecewise linear mixed model with HIV DNA levels as a quantitative variable. A sample size of at least 158 individuals in total would be enough to achieve 80% power to detect the effect of HIV DNA levels, as a quantitative variable, on the two slopes, at the 0.05 alpha level (1000 simulations).

### 2.3. Sample Measurements

Quantification of total HIV DNA per PBMC was determined in one cryopreserved PBMC sample taken before the initiation of cART. Total DNA was isolated from PBMCs using QIAamp DNA blood mini kit (Qiagen GmbH, Hilden, Germany) according to the manufacturer’s recommendations; extracted DNA was eluted with 100 μL DNase-free water. HIV DNA was quantified by real time PCR (RTD-PCR) using molecular beacons as a detection system to quantify all HIV DNA forms (single-stranded DNA and dsDNA forms and integrated and unintegrated linear or circular forms, and the limit of detection was 12 c/mL). HIV DNA was quantified in parallel with CCR5 as a reference gene and reported values were numbers of HIV DNA copies/10^6^ PBMCs in a LightCycler 1.0 instrument [14]. HIV DNA and CCR5 were quantified in separate reactions using primers, molecular beacons and thermocycling conditions as described previously [15]. External standards were prepared from ccR5 and HIV amplicons and they were calibrated according to our previous quantifications [15].

HIV RNA was measured in plasma in the same blood samples used for DNA measurements using Chiron b-DNA 3.0 (Chiron, Emeryville, CA, USA). CD4 counts were measured in whole blood samples using flow cytometry immunophenotyping using FACSCalibur (BD Biosciences, Franklin Lakes, NJ, USA).

### 2.4. Statistical Analysis

Piecewise-linear mixed models were used to describe the changes of CD4 cell counts (compared to baseline levels) over time and to assess the impact of baseline HIV DNA levels, adjusting for potential confounders. Models allowed for different rates of CD4 changes during the first 3 months of antiretroviral treatment and after the 3rd month and included random effects for the intercept and both slopes.

## 3. Results

### 3.1. Participant Characteristics

One hundred forty-eight participants fulfilled the inclusion criteria of the study. The main demographic and clinical characteristics are summarized in Table 1.

Initial HIV DNA levels were greater in the people who inject drugs (PWID), the heterosexual, and the non-Greek population (58.5%, 72%, and 70.4% above median). Participants were stratified according to HIV DNA level (stratum A, <2.58 log_10_ copies/10^6^ PBMCs; stratum B, ≥2.58 log_10_ copies/10^6^ PBMCs). HIV DNA level was below the detection limit of 12 copies/mL for 12 study participants, and it was replaced with 6 copies/mL. Median (interquartile range, IQR) HIV RNA was 4.7 (4.3, 5.2) log_10_ copies/mL. HIV DNA level was significantly correlated with HIV RNA level (Spearman’s ρ = 0.208, *p* = 0.011) baseline (Spearman’s ρ = −0.179, *p* = 0.029) and nadir CD4 T cell count (Spearman’s ρ = −0.185, *p* = 0.025) (Figure 1). Based on the combination of HIV DNA and RNA levels we identified four groups low/low (n = 41), low/high (n = 33), high/low (n = 33), and high/high (n = 41) with “low” and “high” denoting levels below or above median, respectively.

The median (IQR) time for the first CD4 measurement after cART initiation was 3 (1.9, 4.2) months and the median (IQR) interval between measurements was 6.1 (5.2, 7.4) months. The median (IQR) number of CD4 T-cell measurements per participant was 9 (6, 12) with no difference between those with below and those with above median HIV DNA. The median (IQR) CD4 count at baseline was 298 (154, 441.5) cells/μL. CD4 count continued to increase over the entire 2 years period after individuals started cART, although the steepest increase occurred in the first three months (Appendix A).

### 3.2. Univariable Analysis

In univariable analysis, baseline HIV RNA level was associated with the change in CD4 count especially during the first 3 months (60.8 cells/μL/month; 95% CI: 49, 72.5, *p* < 0.001 for those with HIV RNA below median) after cART initiation, with the higher viral load associated with a greater increase in CD4 count (difference between those with HIV RNA above vs. below median 15.1 cells/μL/month; 95% CI: −1.4, 31.5; *p* = 0.073) (Appendix A). The same observation applied for HIV DNA level with the higher level being associated with marginally non-significant faster CD4 increases after the 3rd month of cART (difference between those with HIV DNA above vs. below median 1.2 cells/μL/month; 95% CI: −0.1, 2.6; *p* = 0.071) (Appendix A). When combining the effects of HIV RNA and HIV DNA, it seemed that higher levels in both predicted the rate of CD4 increase after the 3rd month of cART (difference between those with both HIV RNA and HIV DNA above median vs. both below median 2.1 cells/μL/month; 95% CI: 0.3–4; *p* = 0.024) (Table 2 and Figure 2).

Baseline CD4 or gender did not predict the rate of CD4 increase. There was a tendency though, for individuals starting cART with very low (i.e., <100 CD4 cells/μL; n = 23) or very high (i.e., ≥500 CD4 cells/μL; n = 22) CD4 cell counts to have lower CD4 gains during the first 3 months of treatment. Other variables associated with the change in CD4 cell count included age at baseline (inverse correlation in those aged more than 50 years old) and risk group. Especially in PWIDs, CD4 increases during the first 3 months of treatment were significantly slower compared to MSM (difference 29.8 cells/μL/month, 95% CI 10.4, 49.2, *p* = 0.003) (Appendix A). Examining our entire population sample, no association was found between the type of cART or AIDS before cART initiation and CD4 slope change.

### 3.3. Multivariable Analysis

In multivariable analysis, baseline CD4 cell count was an independent factor for CD4 changes after cART initiation with more pronounced increases in those with lower CD4 T cell values. In the subgroup of patients with CD4 cell count below median (i.e., <298 cells/μL), analyzing the factors that influence the CD4 cell change after cART initiation, intravenous drug use and age above 50 years old inversely affected the CD4 change. On the contrary, patients with high HIV RNA and high HIV DNA had a significantly faster increase in CD4 cell count after 3 months on cART. In this subgroup of participants, a similar association was observed in individuals who received INSTIs (Table 3).

In the subgroup of patients with CD4 cell count above the median (i.e., ≥298 cells/μL), intravenous drug use inversely affected the increase of CD4 cells in the first three months after cART initiation, and patients with higher level of both HIV DNA and HIV RNA had a significantly higher increase in CD4 cell count during the first three months on cART (Table 4).

Suboptimal CD4 response (<100 CD4 cells/μL gain at 6 months or <500 cells/μL at 24 months after cART initiation) was independently associated with lower baseline CD4 cell count, intravenous drug use, and age 50 years or older. Individuals on boosted PIs had significantly lower risk for a suboptimal CD4 recovery at 6 months (Appendix A).

During follow-up, 10 patients presented with immune reconstitution inflammatory syndrome (IRIS) at a median time (IQR) of 73 (18, 87) days from treatment onset. The emergence of IRIS was associated with lower initial CD4 counts [37 (20, 175) vs. 310 (185, 494), *p* < 0.001)], but not with initial HIV DNA or RNA levels.

Over a median follow-up of 4.1 (3.2, 6) years with 10 participants being lost, five PLWH (5/138) died with no correlation to initial HIV DNA levels.

## 4. Discussion

In our study, we attempted to investigate if measuring HIV DNA levels was associated with immune reconstitution. Initial HIV DNA differences between transmission risk groups possibly reflect the higher percentage of late-presentation (i.e., receiving an HIV infection diagnosis with either an AIDS-defining condition or a CD4 count <250 cell/μL at presentation) in PWIDs and heterosexuals as opposed to MSMs [16], although the HIV DNA set point is known to be established early in the natural course of infection [17]. It is of note that in published literature, non-Caucasian populations and PWID had lower HIV DNA levels, a fact attributed at least partially to different viral subtypes in these transmission groups [18]. In our data we found a significant association between initial HIV DNA levels and initial RNA levels and CD4 counts, and nadir CD counts, albeit the degree of the effect seems small. It seems that the DNA set point as measured with our method cannot be predicted entirely based on the RNA set point or the CD4 counts but there are other factors in play as well.

It appears that HIV DNA can serve as a complementary biomarker to HIV RNA measurements, and the combination of the two can successfully predict the rate with which CD4 counts will improve, particularly after the first three months of treatment. Overall increases in CD4 cell count were biphasic with a steep initial increase followed by a milder long-term increase. The CD4 count continued to rise for at least two years after cART initiation, although the steepest slope was observed during the first three months. We detected small differences based on the initial CD4 counts with regards to the role that the HIV DNA levels play. Among those with high CD4 counts, the combination of high HIV DNA and RNA levels was not associated with immunological improvement past the first trimester, but in the first three months there was a trend towards such a finding, marginally missing statistical significance. On the other hand, among those with low CD4 counts, in the first trimester there was no association between immunological improvement (as evidenced by a rise in CD4 counts) and HIV DNA and RNA levels, but we observed a statistically significant improvement over the following months in the group of high/high DNA/RNA levels as opposed to the low/low group.

The overall positive effect of the combined high RNA/DNA values on CD4 increase is of particular interest and to our knowledge a novel finding. It is suggested that the initiation of cART with the associated drop in viral loads would account for this increase, whereas in patients with lower viral loads, the immunological recovery of the host is less pronounced. The lack of clear association with RNA levels alone, implies that the viral reservoir itself, as reflected on HIV DNA levels, is at least partially responsible for the immune system suppression. Using a combination of the two markers seems to offer a more concise picture of the immune system.

Knowing the rate of immune reconstitution can help the clinician anticipate treatment related adverse events, i.e., the emergence of IRIS. Predicting IRIS with the use of biomarkers such as HIV DNA can help prevent it. Even though we found no correlation between IRIS and the initial levels of HIV DNA or RNA in our study, it was not designed with a sample large enough to be able to make such a distinction. It is evident in the literature that both high levels of viral loads and their associated drop after cART initiation, especially if more than 2 log_10_, is associated with IRIS [19,20]. On the other hand, IRIS has been reported in patients before even the increase in CD4 counts appeared, after cART initiation. Our study shows that in patients with very high viral DNA and RNA loads, the associated, anticipated steep increase in CD4 is the missing link for the emergence of IRIS. Moreover, the association of HIV DNA levels and other events, i.e., the emergence of AIDS and all-cause mortality, has been demonstrated in meta-analysis [21].

Of particular interest, we note the finding of the positive association of INSTI on changes of CD4 count over time in the low CD4 subgroup. There have been studies linking the emergence of IRIS to the use of INSTIs [22,23,24]. It has also been demonstrated that in virologically suppressed patients on PI or NNRTI based regimens, that switching to raltegravir causes a decline in HIV DNA levels at 48 weeks [25,26]. Given that lower HIV DNA levels in chronically infected treated patients are associated with greater CD4 counts, our results come to bridge those two observations. In contrast, a pilot study comparing a standard regimen of TDF/FTC/EFV to a regimen further intensified with a 4th and 5th agent of raltegravir and maraviroc, respectively, in acutely infected patients, failed to show a more drastic decline in either HIV DNA levels or a rise in CD4 counts. The study noted that the rise in CD4 counts was more pronounced closer to cART initiation in accordance with our own results as well [27].

A lot is known in regards to the effect of HIV DNA levels on virological suppression after treatment, but few studies have examined their role on the rate of immune reconstitution. It has been demonstrated that initial higher HIV DNA levels were associated with the presence of undetectable residual viremia in patients after 4 years of treatment [28]. In a Spanish study of 115 patients, investigators measured total HIV DNA levels pre-cART and at 6 weeks after treatment initiation. They found that HIV DNA levels were able to predict virologic failure, independent of and with a stronger association than CD4 counts, and RNA levels measured at the same time. The mean time to virological failure was also inversely associated with HIV DNA levels [29].

On the other hand, Poizot-Martin et al. reported that no association was identified between HIV DNA levels at diagnosis and either CD4 cell count increase after treatment initiation or immune activation as measured by the proportion of CD8+CD38+ and CD8+DR+ T cells [30]. A recent multicenter study evaluated virologically suppressed children and identified three different subphenotypes. The group with higher HIV DNA levels also had shorter telomere lengths, greater percentage of immunosenescent CD8 cells, lower T-cell receptor excision circles (TRECs) and higher vascular cell adhesion molecule (VCAM) levels [31].

The present study has certain limitations. Our sample is derived from a single-center cohort of patients and thus it is possibly not applicable to different populations. Serial measurements of viral RNA and DNA would contribute to our analysis, yet the lack of them does not diminish the importance of our findings. Even though the case is not so in our set of data, it is possible that initial DNA or RNA levels can direct immune response as well, given that these initial levels reflect the degree of initial immune damage. Further studies would be required to investigate the interplay of these markers and eventual immune reconstitution. Finally, although HIV DNA per PBMC is admittedly a good surrogate marker for the entire viral reservoir, tissue DNA measurements would offer additional insight into the interactions investigated in our study.

## 5. Conclusions

HIV DNA affects the CD4 slope in individuals with DNA and RNA levels above the median having a greater increase in CD4 cell count, 3 months after cART initiation. Subjects with a higher HIV RNA, higher HIV DNA or lower CD4 count at baseline had the highest increase in their CD4 count after starting HAART. In successfully treated patients, pre-cART DNA and RNA levels are predictors of immune reconstitution.

## Figures and Tables

**Figure 1 microorganisms-11-01510-f001:**
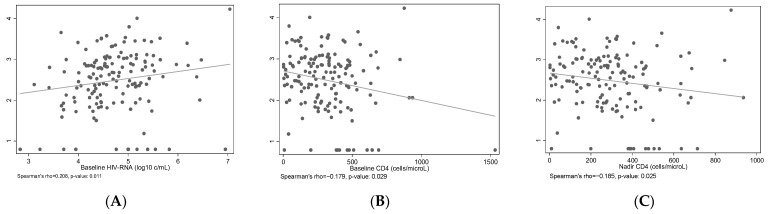
Scatterplots showing the association of log_10_ HIV DNA (/10^6^ PBMCs) with baseline HIV RNA (log_10_ copies/mL) (**A**), baseline (**B**) and nadir CD4 counts (cells/μL) (**C**). Grey straight lines correspond to linear regression fitted lines.

**Figure 2 microorganisms-11-01510-f002:**
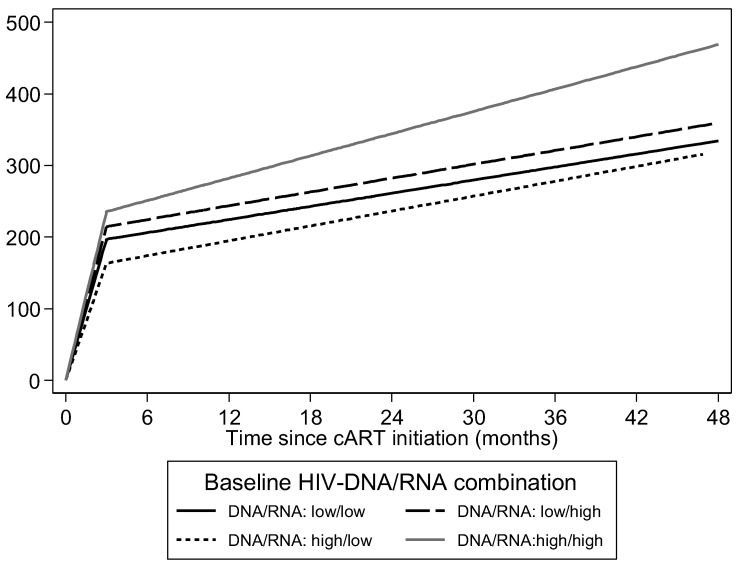
Estimated average change of CD4 cell count levels over time, after cART initiation. Estimates shown for combinations of HIV-DNA and HIV-RNA levels (above or below median denoted as “high” or “low”, respectively) at cART initiation. Estimates are based on an unadjusted mixed model.

**Table 1 microorganisms-11-01510-t001:** Demographic and clinical characteristics of the study participants by HIV DNA levels and overall.

	HIV DNA			
	Below Median * (n = 74)	Above Median * (n = 74)	Overall (n = 148)	
	N(%) or Median (IQR)	N(%) or Median (IQR)	N(%) or Median (IQR)	*p*-Value
Sex				0.355
* Male*	65 (51.6)	61 (48.4)	126 (100.0)	
* Female*	9 (40.9)	13 (59.1)	22 (100.0)	
Risk group				**0.007**
* MSM*	50 (61.0)	32 (39.0)	82 (100.0)	
* PWID*	17 (41.5)	24 (58.5)	41 (100.0)	
* Heterosexual*	7 (28.0)	18 (72.0)	25 (100.0)	
Risk group & Sex				**0.024**
* MSM*	50 (61.0)	32 (39.0)	82 (100.0)	
* PWID-Male*	13 (38.2)	21 (61.8)	34 (100.0)	
* PWID-Female*	4 (57.1)	3 (42.9)	7 (100.0)	
* Heterosexual-Male*	2 (20.0)	8 (80.0)	10 (100.0)	
* Heterosexual-Female*	5 (33.3)	10 (66.7)	15 (100.0)	
Origin				**0.019**
* Greece*	66 (54.5)	55 (45.5)	121 (100.0)	
* Other*	8 (29.6)	19 (70.4)	27 (100.0)	
Median Age at cART initiation (years)	35 (30, 45)	36 (31, 43)	36 (30, 43)	0.765
Primary/recent infection				>0.999
* No*	67 (50.0)	67 (50.0)	134 (100.0)	
* Yes*	7 (50.0)	7 (50.0)	14 (100.0)	
Type of cART				0.557
* NNRTI*	49 (52.1)	45 (47.9)	94 (100.0)	
* Boosted PI*	19 (50.0)	19 (50.0)	38 (100.0)	
* INSTI.*	6 (37.5)	10 (62.5)	16 (100.0)	
AIDS before cART				0.092
* No*	64 (47.8)	70 (52.2)	134 (100.0)	
* Yes*	10 (71.4)	4 (28.6)	14 (100.0)	
Death				0.172
* No*	70 (49.0)	73 (51.0)	143 (100.0)	
* Yes*	4 (80.0)	1 (20.0)	5 (100.0)	
Nadir CD4 (cells/microL)				0.245
0–99	11 (47.8)	12 (52.2)	23 (100.0)	
100–199	11 (44.0)	14 (56.0)	25 (100.0)	
200–349	21 (41.2)	30 (58.8)	51 (100.0)	
350–499	19 (63.3)	11 (36.7)	30 (100.0)	
500+	12 (63.2)	7 (36.8)	19 (100.0)	
Baseline CD4 (cells/microL)				0.360
0–99	11 (47.8)	12 (52.2)	23 (100.0)	
100–199	10 (41.7)	14 (58.3)	24 (100.0)	
200–349	18 (41.9)	25 (58.1)	43 (100.0)	
350–499	22 (61.1)	14 (38.9)	36 (100.0)	
500+	13 (59.1)	9 (40.9)	22 (100.0)	
Baseline CD4 (cells/microL)	326 (160, 464)	278 (132, 417)	298 (154, 442)	0.245
Baseline HIV RNA (copies/mL)				0.117
500–9999	15 (75.0)	5 (25.0)	20 (100.0)	
10,000–49,999	26 (44.8)	32 (55.2)	58 (100.0)	
50,000–99,999	9 (45.0)	11 (55.0)	20 (100.0)	
100,000+	24 (48.0)	26 (52.0)	50 (100.0)	
Baseline HIV RNA (log_10_ c/mL)	4.5 (4.2, 5.1)	4.7 (4.5, 5.2)	4.7 (4.3, 5.2)	0.051
Log_10_ HIV DNA (/10^6^ PBMCs)	2.0 (1.7, 2.4)	3.0 (2.8, 3.3)	2.6 (2.0, 3.0)	**<0.001**
1st HIV+ to cART (months)	3.1 (1.1, 9.3)	2.7 (1.0, 9.2)	2.9 (1.0, 9.2)	0.809
1st HIV+ to HIV DNA sample (months)	0.4 (0.2, 0.9)	0.5 (0.2, 1.3)	0.4 (0.2, 1.1)	0.446

Percentages as presented as part of the overall population for each variable. Non-parametric tests (Mann-Whitney U-test) and chi-square tests have been used for continuous and categorical variables, respectively. Findings with statistical significance (*p* < 0.05) are presented in bold. * HIV DNA median value: 2.58 log_10_ copies/10^6^ PBMCs. MSM: men who have sex with men, PWID: people who inject drugs, cART: combined anti-retrotiviral therapy, NNRTI: non-nucleoside reverse transcriptase inhibitor, PI: protease inhibitor, INSTI: integrase strand transfer inhibitor, PBMC: peripheral blood mononuclear cell.

**Table 2 microorganisms-11-01510-t002:** Results from a mixed model for CD4 cell count changes after cART according to baseline HIV DNA and HIV RNA combinations.

Factor	Estimate (Cells/μL)	95% C.I.	*p*-Value
**CD4 Change 0–3 months** (per month)	65.7	(49.9, 81.4)	**<0.001**
Baseline HIV DNA/RNA combination and CD4 Change 0–3 months interaction			
* DNA*/*RNA: low*/*high* vs. *low*/*low*	5.9	(−17.2, 29.0)	0.617
* DNA*/*RNA: high*/*low* vs. *low*/*low*	−11.2	(−34.8, 12.5)	0.355
* DNA*/*RNA: high*/*high* vs. *low*/*low*	12.8	(−9.3, 34.9)	0.256
**CD4 Change 3+ months** (per month)	3.1	(1.8, 4.3)	**<0.001**
Baseline HIV DNA/RNA combination and CD4 Change 3+ months interaction			
* DNA*/*RNA: low*/*high* vs. *low*/*low*	0.2	(−1.7, 2.0)	0.863
* DNA*/*RNA: high*/*low* vs. *low*/*low*	0.4	(−1.5, 2.3)	0.672
* DNA*/*RNA: high*/*high* vs. *low*/*low*	2.1	(0.3, 4.0)	**0.024**

Note: “high” and “low” denote levels above and below median, respectively, *p*-values in bold indicate statistical significance.

**Table 3 microorganisms-11-01510-t003:** Results from a multivariable mixed model for CD4 changes after cART (subgroup: Baseline CD4 below median, i.e., <298 cells/μL).

Factor	Estimate (Cells/μL)	95% C.I.	*p*-Value
**CD4 Change 0–3 months** (per month)	108.3	(69.7, 146.9)	**<0.001**
Baseline HIV DNA/RNA combination and CD4 Change 0–3 months interaction			
* DNA*/*RNA: low*/*high* vs. *low*/*low*	9.0	(−24.4, 42.4)	0.598
* DNA*/*RNA: high*/*low* vs. *low*/*low*	−7.2	(−42.1, 27.7)	0.685
* DNA*/*RNA: high*/*high* vs. *low*/*low*	−5.8	(−37.9, 26.3)	0.723
Risk group and CD4 Change 0–3 months interaction			
*PWID* vs. *non-PWID*	−37.9	(−65.8, −9.9)	**0.008**
Age at cART initiation and CD4 Change 0–3 months interaction			
30–39 vs. <30	−16.1	(−48.0, 15.8)	0.322
40–49 vs. <30	−23.6	(−57.5, 10.2)	0.171
50+ vs. <30	−39.1	(−77.6, −0.5)	**0.047**
Type of cART and CD4 Change 0–3 months interaction			
* Boosted PI* vs. *NNRTI*	−18.3	(−44.1, 7.4)	0.163
* INSTI* vs. *NNRTI*	−32.2	(−77.0, 12.7)	0.160
**CD4 Change 3+ months** (per month)	3.1	(0.7, 5.5)	**0.013**
Baseline HIV DNA/RNA combination and CD4 Change 3+ months interaction			
* DNA*/*RNA: low*/*high* vs. *low*/*low*	1.0	(−1.1, 3.1)	0.349
* DNA*/*RNA: high*/*low* vs. *low*/*low*	1.2	(−1.2, 3.5)	0.327
* DNA*/*RNA: high*/*high* vs. *low*/*low*	2.5	(0.5, 4.6)	**0.017**
Risk group and CD4 Change 3+ months interaction			
* PWID* vs. *non-PWID*	−0.7	(−2.6, 1.2)	0.480
Age at cART initiation and CD4 Change 3+ months interaction			
30–39 vs. <30	−0.4	(−2.4, 1.6)	0.705
40–49 vs. <30	−2.1	(−4.4, 0.2)	0.076
50+ vs. <30	−2.0	(−4.4, 0.5)	0.119
Type of cART and CD4 Change 3+ months interaction			
* Boosted PI* vs. *NNRTI*	1.1	(−0.5, 2.6)	0.179
* INSTI* vs. *NNRTI*	3.9	(0.3, 7.5)	**0.032**

PWID: people who inject drugs, cART: combined antiretroviral treatment, PI: protease inhibitor, NNRTI: non-nucleoside reverse transcriptase inhibitor, INSTI: integrase strand transfer inhibitor. *p*-values in bold indicate statistical significance.

**Table 4 microorganisms-11-01510-t004:** Results from a multivariable mixed model for CD4 changes after cART (subgroup: Baseline CD4 above median i.e., ≥298 cells/μL).

Factor	Estimate (Cells/μL)	95% C.I.	*p*-Value
**CD4 Change 0–3 months** (per month)	76.1	(49.9, 102.4)	**<0.001**
Baseline HIV DNA/RNA combination and CD4 Change 0–3 months interaction			
* DNA*/*RNA: low*/*high* vs. *low*/*low*	−9.2	(−41.8, 23.4)	0.580
* DNA*/*RNA: high*/*low* vs. *low*/*low*	−7.2	(−37.6, 23.2)	0.642
* DNA*/*RNA: high*/*high* vs. *low*/*low*	30.7	(−1.2, 62.5)	0.059
Risk group and CD4 Change 0–3 months interaction			
*PWID* vs. *non-PWID*	−30.5	(−58.7, −2.3)	**0.034**
Age at cART initiation and CD4 Change 0–3 months interaction			
30–39 vs. <30	−5.6	(−32.0, 20.8)	0.678
40–49 vs. <30	14.5	(−21.9, 50.9)	0.436
50+ vs. <30	−26.6	(−67.9, 14.6)	0.206
Type of cART and CD4 Change 0–3 months interaction			
* Boosted PI* vs. *NNRTI*	13.1	(−27.2, 53.5)	0.523
* INSTI* vs. *NNRTI*	1.5	(−34.5, 37.5)	0.934
**CD4 Change 3+ months** (per month)	2.3	(−0.3, 4.9)	0.089
Baseline HIV DNA/RNA combination and CD4 Change 3+ months interaction			
* DNA*/*RNA: low*/*high* vs. *low*/*low*	−0.2	(−3.5, 3.1)	0.888
* DNA*/*RNA: high*/*low* vs. *low*/*low*	0.2	(−2.8, 3.2)	0.884
* DNA*/*RNA: high*/*high* vs. *low*/*low*	0.7	(−2.7, 4.1)	0.677
Risk group and CD4 Change 3+ months interaction			
* PWID* vs. *non-PWID*	2.0	(−1.2, 5.2)	0.212
Age at cART initiation and CD4 Change 3+ months interaction			
30–39 vs. <30	1.4	(−1.4, 4.2)	0.323
40–49 vs. <30	0.7	(−2.8, 4.3)	0.684
50+ vs. <30	−0.6	(−5.1, 3.9)	0.792
Type of cART and CD4 Change 3+ months interaction			
* Boosted PI* vs. *NNRTI*	−0.5	(−5.1, 4.1)	0.827
* INSTI* vs. *NNRTI*	0.3	(−3.8, 4.4)	0.887

PWID: people who inject drugs, cART: combined antiretroviral treatment, PI: protease inhibitor, NNRTI: non-nucleoside reverse transcriptase inhibitor, INSTI: integrase strand transfer inhibitor. *p*-values in bold indicate statistical significance.

## Data Availability

Data can be made available upon acceptance of the article at the Pergamos repository of the University of Athens.

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
