# Peer review of "HIV RNA/DNA Levels at Diagnosis Can Predict Immune Reconstitution: A Longitudinal Analysis"

_microorganisms, 2023, doi:10.3390/microorganisms11061510_

Round 1
Reviewer 1 Report
This study presents some interesting findings. The most important is that combined RNA and DNA levels pre-ART predict immune reconstitution (i.e. CD4 count increase). The linked finding that HIV RNA level was associated with greater increase in CD4 count during first trimester of treatment and that HIV DNA is associated with greater CD4 increase after the first trimester were not supported with statistical significance (p>0.05).
The basic premise of the finding that HIV DNA levels pre-ART are associated with greater immune reconstitution is problematic - since HIV DNA is associated with lower CD4 count pre-treatment. What is reflected is the finding of many that there is greater immune reconstitution in patients with lower CD4 count at start, provided the immune damage is not too extensive so as to prevent recovery. This recovery or CD4 restoration is a feature of the immune system and the extent of recovery is linked to extent of initial deficiency. In this study it was shown that HIV DNA which is in indirect measure of the reservoir can (when combined with HIV RNA or viral load) predict the extent of immune recovery. But as initial HIV DNA or RNA is associated with the level of initial damage - all these process are interdependent. So. overall this is an interesting novel way of looking at immune reconstitution and its relationship with the reservoir, but future work needs to tease out the interrelatedness of the phenomena.
I feel aspects of the article can be improved (highlighted below). Note that past tense should be used throughout - in the Abstract particularly "is" is used instead of "was". The introduction was fine - but should also include RNA and RNA/DNA combination as this is a key finding - rather than focusing on DNA alone.
The methodology section requires an explanation of HIV RNA (was is standard routine viral load testing, what method or instrumentation, was it done in plasma, was same sample used as for DNA?). Also no mention of CD4 count methodology. The section on assuming rates of CD4 increase need to be expanded - as there is no justification provided. HIV DNA was assumed to be normally distributed - why was this not tested? Statistical analysis does not mention any program the was used and whether a statistician was consulted?
Results. Table 1 summarizes a lot of information and raises questions. Due to the interrelatedness of CD4 recovery and initial deficiency with DNA and RNA levels, it would have been nice to also have had CD4 data presented similarly - i.e. a breakdown of low vs. high (like DNA) and then associations with all factors shown. The p-values are not always clear - for example in risk group and risk group / sex, to what does the p value refer? Is it the combined groups together or is it multi-group statistics?
The presented data on correlations indicates statistical significance, however the r values are fairly low, indicating weak associations - this should be discussed. The figure legends need to be expanded as interpretation can be difficult.
The is discussion of univariable data analysis - is this univariate? I think some of these data can be better expressed.
The key finding of combined DNA and RNA (representing reservoir and plasma virus) being a predictor of immune reconstitution is intriguing, but needs to be better contextualized due to the interrelatedness. This also influences the suggestions e.g. of using viral DNA to predict IRIS.
English is generally good. Check univariable vs. univariate. Better figure legends required. Check use of tenses.
Reviewer 2 Report
First of all, congratulations on the research carried out. The authors presented results that investigated HIV RNA/DNA levels as possible markers of immune reconstitution before starting cART and after starting cART, providing important information in predicting immune restoration, helping to manage the therapeutic strategy used.
Author Response
We appreciate all the time and effort put into reviewing our manuscript. Thank you for the encouragement and the kind words!
Reviewer 3 Report
Basoulis et al. did a retrospective study on a cohort of people living with HIV in Greece (n=148). The overall goal was to test one time point of PBMCs for cell-associated DNA/RNA HIV virus before antiretroviral treatment was started and look into potential correlations with immune reconstitution. The study showed that high level of HIV RNA and DNA was associated with a greater increase in CD4 counts after ART was started. The RNA association was seen during the first trimester on ART; the DNA association was seen after the first trimester. The study addressed a relevant topic, has some relevant data, and the results are in line with similar studies in the field. The study has some limitations (only one sample of PBMCs was tested per individual; the study only used PBMCs so reservoirs in tissues were not addressed) but the authors properly stated them in their discussion, so in that regard I found the manuscript balanced.
I have some comments/suggestions to the authors:
· My main concern with this study is that it was not clear to me how the CD4 counts were obtained for the participants in the study. When reading the material and methods section, and more specifically the section entitled “2.2 sample size estimation”, it seems to me the CD4 counts were somehow inferred by some statistical test (simulation). If so, I am concerned about the robustness of that approach, more specially when used to infer correlations with such estimated data. I think it would be helpful if the authors could properly clarify this point and the 2.2 section.
I also have some minor comments:
· Line 248: I found confusing the expression “late-presentation”. Please clarify what was meant here.
· Figure 1: I suggest adding a title to the y-axis of these graphs.
· Line 90: could the authors clarify what is AIDS status? Would it literally be whether the individuals had developed AIDS or not? Was the usual criteria of <200 CD4 cells per µL used?
· Some tables are missing the bold format in the significant p-values, please update for consistency.
· Authors used CCR5 as a control for their PCRs. Out of curiosity, could they tell from their tests any potential frequency of the delta 32 allele?
· Lines 304-305: something seems to be missing in these lines. Please correct.
Manuscript could benefit from some minor edits, some of them are reflected in my comments to the authors.
Author Response
We appreciate all the time and effort put into reviewing our manuscript and suggesting ways to improve it. We are grateful for your insights. Replies to specific points raised by the reviewers follow below.
REVIEWER 3
Basoulis et al. did a retrospective study on a cohort of people living with HIV in Greece (n=148). The overall goal was to test one time point of PBMCs for cell-associated DNA/RNA HIV virus before antiretroviral treatment was started and look into potential correlations with immune reconstitution. The study showed that high level of HIV RNA and DNA was associated with a greater increase in CD4 counts after ART was started. The RNA association was seen during the first trimester on ART; the DNA association was seen after the first trimester. The study addressed a relevant topic, has some relevant data, and the results are in line with similar studies in the field. The study has some limitations (only one sample of PBMCs was tested per individual; the study only used PBMCs so reservoirs in tissues were not addressed) but the authors properly stated them in their discussion, so in that regard I found the manuscript balanced.
I have some comments/suggestions to the authors:
- My main concern with this study is that it was not clear to me how the CD4 counts were obtained for the participants in the study. When reading the material and methods section, and more specifically the section entitled “2.2 sample size estimation”, it seems to me the CD4 counts were somehow inferred by some statistical test (simulation). If so, I am concerned about the robustness of that approach, more specially when used to infer correlations with such estimated data. I think it would be helpful if the authors could properly clarify this point and the 2.2 section.
CD4 count estimates were used only for the sample size estimation. CD4 counts for study participants were measured using flow cytometry and we have added this now in the methods section, per your suggestion.
I also have some minor comments:
- Line 248: I found confusing the expression “late-presentation”. Please clarify what was meant here
- Added a small explanation in parenthesis.
- Figure 1: I suggest adding a title to the y-axis of these graphs.
- In our manuscript copy, there is a y axis labeled Log10 HIV DNA. Perhaps there is some issue with the upload, we will see if there needs to be some correction with it.
- Line 90: could the authors clarify what is AIDS status? Would it literally be whether the individuals had developed AIDS or not? Was the usual criteria of <200 CD4 cells per µL used?
- Standard definition was used, and we added it in parenthesis for clarity.
- Some tables are missing the bold format in the significant p-values, please update for consistency.
- Fixed, thank you.
- Authors used CCR5 as a control for their PCRs. Out of curiosity, could they tell from their tests any potential frequency of the delta 32 allele?
- The real-time PCR assay was not designed to quantify the frequency of the different CCR5 alleles since the targeted region was adjacent to the Δ32 deletion of the CCR5 gene and therefore can bind to both the wild-type CCR5 and mutant CCR5Δ32 alleles, as mentioned in the cited paper (Beloukas et al, 2009)
- Lines 304-305: something seems to be missing in these lines. Please correct.
- Rephrased for clarity.
please see attachment
